# Antioxidative Role of *Hygrophila erecta* (Brum. F.) Hochr. on UV-Induced Photoaging of Dermal Fibroblasts and Melanoma Cells

**DOI:** 10.3390/antiox11071317

**Published:** 2022-07-02

**Authors:** Su Jin Lee, Ji Eun Kim, Yun Ju Choi, You Jeong Jin, Yu Jeong Roh, A Yun Seol, Hee Jin Song, So Hae Park, Md. Salah Uddin, Sang Woo Lee, Dae Youn Hwang

**Affiliations:** 1Department of Biomaterials Science (BK21 FOUR Program), Life and Industry Convergence Research Institute, College of Natural Resources and Life Science, Pusan National University, Miryang 50463, Korea; nuit4510@naver.com (S.J.L.); prettyjiunx@naver.com (J.E.K.); poiu335@naver.com (Y.J.C.); hjinyuu1@naver.com (Y.J.J.); buzyu99@naver.com (Y.J.R.); a990609@naver.com (A.Y.S.); hejin1544@naver.com (H.J.S.); sohaehw@pusan.ac.kr (S.H.P.); 2Ethnobotanical Database of Bangladesh, Tejgaon, Dhaka 1208, Bangladesh; plantsofbd@gmail.com; 3International Biological Material Research Center, Korea Research Institute of Bioscience and Biotechnology, Daejeon 34141, Korea; ethnolee@kribb.re.kr; 4Longevity & Wellbeing Research Center and Laboratory Animals Resources Center, College of Natural Resources and Life Science, Pusan National University, Miryang 50463, Korea

**Keywords:** *Hygrophila erecta*, photoaging, ROS, antioxidant, inflammation, ECM, melanin

## Abstract

Antioxidants are an important strategy for treating photoaging because excessive reactive oxygen species (ROS) are produced during UV irradiation. The therapeutic effects of methanol extracts of *Hygrophila erecta* (Brum. F.) Hochr. (MEH) against UV-induced photoaging were examined by monitoring the changes in the antioxidant defense system, apoptosis, extracellular matrix (ECM) modulation, inflammatory response, and melanin synthesis in normal human dermal fibroblast (NHDF) cells and melanoma B16F1 cells. Four bioactive compounds, including 4-methoxycinnamic acid, 4-methoxybenzoic acid, methyl linoleate, and asterriquinone C-1, were detected in MEH, while the DPPH free radical scavenging activity was IC_50_ = 7.6769 µg/mL. UV-induced an increase in the intracellular ROS generation, NO concentration, SOD activity and expression, and Nrf2 expression were prevented with the MEH treatment. Significant decreases in the number of apoptotic cells, the ratio of Bax/Bcl-2, and cleaved Cas-3/Cas-3 were observed in MEH-treated NHDF cells. The MEH treatment induced the significant prevention of ECM disruption and suppressed the COX-2-induced iNOS mediated pathway, expression of inflammatory cytokines, and inflammasome activation. Finally, the expression of the melanin synthesis-involved genes and tyrosinase activity decreased significantly in the α-melanocyte-stimulating hormone (MSH)-stimulated B16F1 cells after the MEH treatment. MEH may have an antioxidative role against UV-induced photoaging by suppressing ROS-induced cellular damage.

## 1. Introduction

Unlike chronological aging, photoaging is defined as premature skin aging caused by repeated exposure to UV, which accompanies thickening of the skin, generation of deep wrinkles, and pigmentation [1]. During this pathological condition, reactive oxygen species (ROS) are overproduced. They are a significant cause of covalent adduct formation and oxidation of various cellular components, including DNA, proteins, and lipids, disrupting the redox balance in antioxidative defense mechanisms [2,3]. An enhanced level of ROS mediates immunosuppression and photoaging [4]. In the immunosuppression response, it activates the mitogen-activated protein kinase (MAPK) signaling pathway and stimulates the translocation of activated nuclear factor kappa light chain enhancer of activated B cells (NF-κB) and the transcription factor activator proteins 1 (AP-1) into the nucleus, which promotes the secretion of inflammatory cytokines [5,6]. The inflammatory cytokines promote the translocation of the Bax protein, which triggers the mitochondria to release cytochrome C and induce apoptosis. ROS can damage DNA indirectly by generating peroxided bases, such as 8-oxoguanine [7]. In the photoaging response, overproduced ROS stimulates the increase in MMP and elastin expression and the decrease in protocollagen synthesis by regulating the transforming growth factor-β (TGF-β) and AP-1 transcription factors [4,8]. These alternative regulations lead to an increase in collagen breakdown and elastin accumulation and a decrease in collagen production, resulting in the formation of wrinkles, dyspigmentation, and dryness in the skin [4,9].

Antioxidants are substances that can prevent cellular damage caused by free radicals. They are regarded as an excellent strategy to improve and alleviate the symptoms of photoaging because of the close connection between UV-induced photoaging and ROS [10]. Among the various antioxidants, retinoids, a class of compounds derived from vitamin A, promote cell turnover and improve skin hyperpigmentation by inhibiting the UV-induced accumulation of melanin and suppressing skin roughening and the formation of wrinkles by stimulating collagen production [10,11,12]. Moreover, fluorouracil cream, which blocks the growth of abnormal cells and is used in body areas with a high density of actinic keratoses, reduces the number of keratoses by approximately 70% and decreases the epidermal roughness associated with photoaging [13]. On the other hand, these compounds can have side effects, such as skin redness, scaling, pruritus, and dryness on the skin phenotypes, while it induces hyperplasia of the epidermis and skin thickening on the histological structures [14].

To overcome the shortcomings of these chemical compounds for a photoaging treatment, the identification of a novel natural product with fewer side effects has attracted considerable attention. Among them, *Ranunculus bulumei*, wheat extract oil, and *Ulmus macrocarpa* Hance. have attracted attention as a photoaging treatment due to their high antioxidant ability without significant side effects. The methanol extract of *Ranunculus bulumei* (Rb-ME) has abundant flavonoids, such as quercetin, luteolin, and kaempferol. With its high antioxidant effect, Rb-ME suppresses MMP-9 and COX-2 expression and increases the release of hyaluronan (HA) synthase by suppressing the UV-induced activation of p38. [15]. Wheat extract oil increases the expression of procollagen Type 1 and HA in HaCaT cells irradiated with UVB [16]. This oil also inhibits melanin production in α-MSH-treated B16F10 cells by suppressing the expression and activity of tyrosinase [16]. Moreover, the *Ulmus macrocarpa* Hance. extract (UMH) inhibits ROS production by increasing the expression of antioxidant enzymes in H_2_O_2_-treated HDF cells and inhibits wrinkle formation and skin thickening in UVB-irradiated hairless mice. [17]. However, no study has examined the anti-photoaging effects and the mechanism of action of *H. erecta* extract in skin fibroblasts and melanoma cells.

This study examined the therapeutic effects and mechanism of action of MEH on improving photoaging in UVB-irradiated skin fibroblasts and α-MSH-stimulated melanoma cells.

## 2. Materials and Methods

### 2.1. Preparation and Extraction of MEH

A lyophilized sample of MEH (FBM 223-006) was provided by the International Biological Material Research Center of the Korea Research Institutes of Bioscience and Biotechnology (Daejeon, Republic of Korea). Briefly, the dried leaf and stem powder of *H. erecta* (Brum. F.) Hochr. was mixed with methanol at a fixed liquor ratio (1:10, powder: methanol). The mixture was subjected repetitively to the following steps: sonication for 15 min, followed by incubation for 2 h 10 times per day, for three days. The mixture was then filtered through a 0.4 µm pore size filter. This extract was then concentrated using a rotary evaporator (N = 1000 SWD, EYELA, Bohemia, NY, USA) and lyophilized using a speed vacuum concentrator (Biotron Co., Marysville, WA, USA). The final MEH sample was dissolved in dimethyl sulfoxide (DMSO, Duchefa Biochemie, Haarlem, The Netherlands) to the appropriate concentrations to treat both cell lines.

### 2.2. Liquid Chromatography—Mass Spectrometry (LC–MS) Analysis

LC–MS analysis was performed using high-performance–liquid chromatography (HPLC) (Agilent 1290 Infinity, Agilent Technologies, Waldbronn, Germany), coupled with a hybrid quadrupole time-of-flight (Q-TOF) mass spectrometer (6530, Agilent Technologies). Their signals were measured on a mass spectrometer operating in negative ionization mode. An chromatographic separation was performed with an ACQUITY UPLC HSS T3 Column (2.1 mm × 100 mm, 1.8 μm) (Waters, Milford, MA, USA) under the specific conditions: 10 μL of injection volume, 0.3 mL/min in flow rate, 0.1% formic acid-water of mobile phase A, and 0.1% formic acid-acetonitrile of mobile phase B. MS were detected under the following conditions: 300 °C of gas temperature, 9 L/min of gas flow, 45 psig of nebulizer pressure, 350 °C of sheath temperature, 11 L/min of sheath gas flow, 4000 V of VCap and 175 V of fragmentor voltage. All acquisition and data analyses were controlled using MassHunter software (version B. 0600, Agilent Technologies). Bioactive compounds with high concentrations were quantified using the base peaks, which were selected for the quantification of peak area and retention time. The percentage peak area of the target component peak was calculated as the ratio of the area of the target component peak to the total area of all detected peaks to analyze quantity in MEH using LC-MS. This method is used to determine an approximate concentration of bioactive compounds in MEH.

### 2.3. Free Radical Scavenging Activity

The 2,2-diphenyl-1-picrylhydrazyl (DPPH) radical scavenging activity was determined using the method described elsewhere [18]. Briefly, powdered MEH was dissolved in 50% EtOH (100 µL) to 12 different MEH concentrations (1 to 2000 µg/mL), which were then mixed with 100 µL of 0.1 mM DPPH (Sigma–Aldrich Co., St. Louis, MO, USA) in a 95% ethanol solution, or with 100 µL of a 95% ethanol solution (control), followed by incubation at room temperature for 30 min. The absorbance at 517 nm was measured using a Versa Max plate reader (Molecular Devices, Sunnyvale, CA, USA). The DPPH radical scavenging activity of MEH was represented as the IC_50_ value.

### 2.4. Cell Viability Analysis

NHDF (Normal human dermal fibroblasts) and B16F1 murine melanoma cells were provided by the ATCC (Manassas, VA, USA). They were cultured in Dulbecco’s Modified Eagle’s Medium (DMEM, Welgene, Gyeongsan-si, Korea) containing 10% fetal bovine serum (FBS), 2 mM glutamine, 100 U/mL of penicillin, and 100 μg/mL streptomycin, and incubated in humidified 5% CO_2_ and 95% air at 37 °C.

The viability of NHDF and B16F1 cells was determined using an MTT (3-[4,5-dimethylthiazol-2-yl]-2,5-diphenyltetrazolium bromide) assay (Sigma-Aldrich Co.). Briefly, both cell types were seeded at 3 × 10^4^ cells in 200 μL DMEM and cultured for 24 h at 37 °C in a 5% CO_2_ incubator. After reaching 70–80% confluence, NHDF cells were exposed to varying intensities of UVB (50, 55, 60, 65, and 70 mJ/cm^2^) or MEH (50, 100, 200, and 400 μg/mL) for 24 h to determine their optimal dosage. Viability was progressively decreased by UVB radiation between 45 mJ/cm^2^ and 70 mJ/cm^2^, with 70% cell viability observed at 55 mJ/cm^2^ (Appendix A). Cell viability was maintained at 200 μg/mL MEH for 24 h (Appendix A). Based on these results, the optimal conditions of UV radiation and MEH dosage for the treatment of NHDF cells were determined at 55 mJ/cm^2^ and 50, 100, and 200 μg/mL, respectively. A similar response to MEH on cell viability was measured in B16F1 cells; the optimal dosage of MEH for B16F1 cells was determined at 50, 100, and 200 μg/mL MEH (Appendix A).

UV irradiation was performed using a TL 20W/12 RS SLV/25 UVB Broadband TL lamp (Philips, Amsterdam, The Netherlands). UV radiation intensities (mW/cm^2^) were measured at 30 cm from a light source using a UVP UVX^TM^ Digital Radiometer (Analytik Jena US LLC, Upland, CA, USA). This value was converted to mJ/cm^2^ using the following formula:mJ = mW × s (second)(1)

The anti-apoptotic effects, ECM modulation, and anti-inflammatory response were measured to determine the therapeutic effects of MEH on the antioxidant defense system. NHDF cells were divided into the following six groups: UV + Vehicle (DMSO)-treated group, UV + 400 μg/mL MED (Methanol extracts of *Dipterocarpus tuberculatus*)-treated group (UV + MED), UV + low concentration MEH-treated group (50 μg/mL; UV + LMEH), UV + medium concentration MEH-treated group (100 μg/mL; UV + MMEH), UV + high concentration MEH-treated group (200 μg/mL; UV + HMEH), and non-irradiated control group (No group). MED was used as a positive control because it has high antioxidant activity and an anti-photoaging effect [19]. Cells were treated with the appropriate dosages of MEH or MED immediately after irradiation with 55 mJ/cm^2^. The harvested cells were used for further analysis.

### 2.5. Apoptotic Cells Analysis

The distribution of apoptotic cells was analyzed using a Muse^TM^ Annexin V and Dead Cell Kit (Millipore Co., Billerica, MA, USA), based on the manufacturer’s protocol. NHDF cells were harvested following irradiation with 55 mJ/cm^2^ of UV and subsequent treatment with MEH or MED for 24 h. The harvested NHDF cells were suspended in DMEM, and 100 μL of this cell suspension (1 × 10^4^ cells/mL) was incubated with Muse^TM^ Annexin V and a Dead Cell Kit (Millipore Co., Billerica, MA, USA) reaction reagent for 20 min at room temperature. Finally, cells were analyzed using a Muse^TM^ Cell Analyzer (Millipore Co., Billerica, MA, USA). After gating based on size, cells were distinguished into four different groups: non-apoptotic cells [Annexin V (-) and 7-AAD (−)], early apoptotic cells [Annexin V (+) and 7-AAD (−)], late apoptotic cells [Annexin V (+) and 7-AAD (+)], and mostly nuclear debris [Annexin V (+) and 7-AAD (+)]. These data are represented as the number of live and apoptotic cells.

### 2.6. Determination of Intracellular ROS Levels

Intracellular ROS levels were determined by staining with 2′,7′-dichlorofluorescein diacetate (DCF-DA; Sigma–Aldrich Co., St. Louis, MO, USA). NHDF cells were harvested after irradiation with 55 mJ/cm^2^ of UV and subsequent treatment with MEH or MED for 24 h. The cells were then incubated with 10 µM DCF-DA for 30 min at 37 °C and washed twice with 1× PBS. The green fluorescence was observed using a fluorescence microscope (Evos m5000, Invitrogen, Waltham, MA, USA).

### 2.7. Nitric Oxide (NO) Concentration

NO concentration was determined from the nitrite concentration because nitrite is considered an indicator of NO production. NHDF cells were seeded at a density of 3 × 10^4^ cells/well and treated with MEH or MED for 24 h after UV irradiation at 55 mJ/cm^2^. After collecting the total supernatants, 100 μL of each supernatant was mixed with 100 μL Griess reagent (Invitrogen, Waltham, MA, USA). The absorbance of this reaction mixture was read at 540 nm using a Versa Max plate reader (Molecular Devices, Sunnyvale, CA, USA).

### 2.8. Superoxide Dismutase (SOD) Activity Analysis

The SOD activity of NHDF cells was measured using an SOD assay kit (Dojindo Molecular Technologies Inc., Rockville, MD, USA) following the manufacturer’s protocol. After the preparation of cell lysates by repetitively freezing and thawing, the supernatants were diluted 1/1, 1/2, 1/22, 1/23, 1/24, 1/25, and 1/26 with a 1 × PBS solution. An appropriate amount of lysate solution (20 μL) was mixed with the WST-1 working solution (200 μL) and the enzyme working solution (20 μL) in 96-well plates. The reaction mixture was incubated at 37 °C for 20 min, and the absorbance was then determined at 450 nm using a spectrophotometer. Finally, SOD activities were determined using the following Equation:SOD activity (inhibition rate %) = [(A blank 1 − A blank 3) − (A sample − A blank 2)]/(A blank 1 − A blank 3) × 100(2)
where A blanks 1, 2, and 3 indicate the absorbance of blanks 1, 2, and 3, respectively, and ‘A sample’ is the sample absorbance. One SOD unit is defined as the amount of the enzyme in the sample (20 µL) that inhibits the reduction reaction of water-soluble tetrazolium salt-1 (WST-1) with superoxide anion by 50%.

### 2.9. Intracellular Elastase Activity Assay

Intracellular elastase inhibition assays were performed based on the modified method described elsewhere [20]. After preparing the cell lysates using 0.2 M Triton-HCl (pH 8.0) buffer containing 0.1% Triton-X, the supernatants were collected as the elastase solution of cells. An appropriate amount of supernatant (100 μL) was mixed with 62.5 mM STANA (N-succinyl-(Ala)3-p-nitroanilide) (2 μL) in a 96-well plate. The control wells contained 100 μL of 0.2 M Triton-HCl (pH 8.0) buffer containing 0.1% Triton-X and 2 μL of 62.5 mM STANA. The reaction mixture was incubated at 37 °C for 1h. Absorbance was then determined at 405 nm using a spectrophotometer. The elastase activity was calculated as follows:Activity (%) = (Absorbance sample/Absorbance control) × 100(3)

### 2.10. Melanin Content Analysis

The cellular melanin content was measured using a slight modification of the methodology described elsewhere [21]. After reaching 70–80% confluence, the B16F1 cells were treated with 1 μM α-MSH (MedChemExpress, Monmouth Junction, NJ, USA) and MEH (50, 100, and 200 μg/mL) for 24 h. The effect of MEH on UV-induced melanin synthesis was examined by dividing the B16F1 cells into the following six groups: α-MSH + Vehicle (DMSO)-treated group, α-MSH + 400 μg/mL MED (α-MSH + MED), α-MSH + low concentration MEH (50 μg/mL; α-MSH + LMEH), α-MSH + medium concentration MEH (100 μg/mL; α-MSH + MMEH), α-MSH + high concentration MEH (200 μg/mL; α-MSH + HMEH), and non-treated control group (No). At the end of the treatment, the cells were washed with 1 × PBS and lysed in 0.2 M Triton-HCl (pH 8.0) buffer containing 0.1% Triton-X. The lysates were then centrifuged for 10 min at 12,000 rpm, and the melanin-containing pellets were dissolved in 1 N NaOH containing 10% DMSO for 30 min at 95 °C. Homogenates (100 μL) were placed in 96-well microplates, and the absorbance was measured at 405 nm.

### 2.11. Intracellular Tyrosinase Activity Assay

B16F1 melanoma cells were seeded at 45 × 10^4^ cells/well in 3 mL medium in 6-well culture plates. The cells were exposed to MEH or MED for 24 h in the presence or absence of 1 μM α-MSH. At the end of the treatment, the cells were washed with 1 × PBS and lysed in 0.2 M Triton-HCl (pH 8.0) buffer containing 0.1% Triton-X. These lysates were then centrifuged for 10 min at 12,000 rpm, and the supernatants were collected. Subsequently, 50 μL of the cell lysate and 50 μL of 5 mM L-DOPA (MedChemExpress, Monmouth Junction, NJ, USA) were added to the well of 96-well plates. The control wells contained 50 μL of lysis buffer and 50 μL of 5 mM L-DOPA. After the plates were incubated at 37 °C for 30 min, absorbance was measured at 475 nm using a Versa max plate reader (Molecular Devices, Sunnyvale, CA, USA).

### 2.12. Western Blot Analysis

Total proteins were obtained from NHDF cells using a Pro-Prep Protein Extraction Solution (Intron Biotechnology Inc., Seongnam, Korea) according to the manufacturer’s protocol. After centrifuging at 13,000 rpm for 5 min, the protein concentrations were determined using a SMART^TM^ Bicinchoninic Acid Protein Assay Kit (Thermo Fisher Scientific Inc., Wilmington, DE, USA). Proteins were separated by 4–20% SDS-PAGE (sodium dodecyl sulfate-polyacrylamide gel electrophoresis) for 2 h, and then transferred to nitrocellulose membranes at 40 V for 2 h. The membranes were then incubated overnight at 4 °C with the following primary antibodies: anti-Bax (Abcam, Cambridge, UK), anti-Bcl2 (Invitrogen, Waltham, MA, USA), anti-caspase3 (Cell Signaling Technology Inc., Danvers, MA, USA), anti-Nrf2 (Abcam, Cambridge, UK), anti-SOD (Abcam), anti-ERK (Cell Signaling Technology), anti-p-ERK (Cell Signaling Technology), anti-JNK (Cell Signaling Technology), anti-p-JNK (Cell Signaling Technology), anti-p38 (Cell Signaling Technology), anti-p-p38 (Cell Signaling Technology), anti-MMP2 (Santa Cruz Bio Technology Inc., CA, USA), anti-MMP9 (Santa Cruz Bio Technology Inc.), anti-ASC (Cell Signaling Technology Inc.), anti-Caspase-1 (Cell Signaling Technology Inc.), cleaved Caspase-1 (Cell Signaling Technology Inc.), anti-NLRP3 (Cell Signaling Technology Inc.), anti-iNOS (Thermo Fisher Scientific Inc.), anti-COX-2 (Cell Signaling Technology Inc.), and anti-β-actin antibody (Cell Signaling Technology Inc.). After removing the non-specific Ab using wash buffer (137 mM NaCl, 2.7 mM KCl, 10 mM Na_2_HPO_4_, and 0.05% Tween 20), each membrane was incubated with horseradish peroxidase (HRP)-conjugated goat anti-rabbit IgG (Invitrogen) at room temperature for 1 h. Finally, chemiluminescence was measured by Fusion Solo-2 (Vilber, San Leandro, Collégien, France) after development using Amersham ECL Select Western Blotting detection reagent (GE Healthcare, Little Chalfont, UK).

### 2.13. Quantitative Real Time-PCR (RT-qPCR) Analysis

Total RNA was purified from NHDF cells using RNAzol (Tet-Test Inc., Friendswood, TX, USA). After determining total RNA concentrations, complementary DNA (cDNA) was synthesized using the Invitrogen Superscript II reverse transcriptase (Thermo Fisher Scientific Inc.). Quantitative PCR(qPCR) was performed using the cDNA template (1 μL) and 2× Power SYBR Green (6 μL; Toyobo Life Science, Osaka, Japan) containing the specific primers. Primer sequences used for target gene expression identification were as follows: HAs1(NM_001523), sense 5′- ATATA GGAAT AACCT CTTGC AGCAG TT -3′ and anti-sense 5′- TGGAG GTGTA CTTGG TAGCA TAACC-3′, HAs2(NM_005328), sense 5′- GCAGC CCATT GAACC AGAGA -3′ and anti-sense 5′- AAGAC TCAGC AGAAC CCAGG AA -3′, HAase1(NM_007312), sense 5′- CTATG ACTTT CTAAG CCCCA ACTAC A -3′ and anti-sense 5′- CCACC CTAGC TGGTC ATTTT G -3′, HAase2(XM_005265524), sense 5′- TCTTC TACCG CGACC GTCTA G -3′ and anti-sense 5′- GCACA CCACC ATGCA CAGA -3′, MITF(XM_006505684), sense 5′- AGCGT GTATT TTCCC CACAG -3′ and anti-sense 5′- TAGCT CCTTA ATGCG GTCGT -3′, TYR(LC259496), sense 5′- GGCCA GCTTT CAGGC AGAGG T -3′ and anti-sense 5′- TGGTG CTTCA TGGGC AAAAT C -3′, TYRP1(XM_006537781), sense 5′- GCTGC AGGAG CCTTC TTTCT C -3′ and anti-sense 5′- AAGAC GCTGC ACTGC TGGTC T -3′, TYRP2(X63349), sense 5′- GGATG ACCGT GAGCA ATGGC C -3′ and anti-sense 5′- CGGTT GTGAC CAATGG GTGCC -3′, TNF-α(NM_013693.3), sense 5′-CCTGT AGCCC ACGTC GTAGC-3′, and anti-sense 5′-TTGAC CTCAG CGCTG ACTTG-3′, IL-6(M26745.1), sense 5′-TTGGG ACTGA TGTTG TTGAC A-3′, and anti-sense 5′-TCATC GCTGT TGATA CAATC AGA-3′, IL-1β(X01450), sense 5′-CAGTT CTGCC ATTGA CCAT-3′, and anti-sense 5′-TCTCA CTGAA ACTCA GCCGT-3′; NF-κB(AY521463), sense 5′-GTAAC AGCAG GACCC AAGGA-3′, and anti-sense 5′-AGCCC CTAAT ACACG CCTCT-3′; β-actin(XM_032887061.1), sense 5′-TGGAA TCCTG TGGCA TCCAT GAAAC-3′, and anti-sense 5′-TAAAA CGCAG CTCAG TAACA GTCCG-3′. qPCR was performed with 40 cycles of denaturation at 95 °C for 15 s, annealing at 57 °C for 15 s, and extension at 72 °C for 45 s. Fluorescence intensities were measured at the end of the extension phase of each cycle. Threshold values for fluorescence intensities were set manually, and the reaction cycles, where the PCR products exceeded these fluorescence intensity thresholds during the exponential phase, were considered to be the threshold cycles (Ct). The expression of Has1, Has2, HAase1, HAase2, MITF, TYR, TYRP1, TYRP2, TNF-α, IL-6, IL-1β, and NF-κB genes were quantified relative to that of the housekeeping gene β-actin, based on a comparison of the Ct values at a constant fluorescence intensity [22].

### 2.14. Statistical Significance Analysis

The statistical difference between No group and treated group was evaluated by one-way analysis of variance (ANOVA), followed by a Tukey’s post hoc test for multiple comparisons using SPSS release 10.10 for Windows (IBM SPSS, SPSS Inc., Chicago, IL, USA). The results are presented as the mean ±SD, and *p* values < 0.05 were considered significant.

## 3. Results

### 3.1. Bioactive Components and Free Radical Scavenging Activity of MEH

LC–MS was performed to identify and characterize the bioactive compounds in MEH. Five compounds, 4-methoxycinnamic acid (7.40%), 4-methoxybenzoic acid (1.82%), methyl linoleate (4.82%), baicalin (2.17%), and asterriquinone C-1 (17.10%), were detected as peaks on the MEH chromatogram (Figure 1a,b). The DPPH radical scavenging activity was measured to predict the antioxidant activity of MEH before mammalian cell analysis. The inhibitory activity against the DPPH radical was increased dose-dependently at 1–1000 µg/mL of MEH, and the IC_50_ value was determined to be 7.6769 µg/mL (Figure 1c). These results suggest that MEH exhibits strong antioxidative activity and has potential for applications as an anti-photoaging complex with high antioxidant activity.

### 3.2. Improvement of the Antioxidant Defense System of UV-Irradiated NHDF Cells by MEH

The intracellular ROS levels, NO concentration, SOD activity, and Nrf2 expression were measured in UV-irradiated NHDF cells after treatment of MEH to determine if the high antioxidant activity of MEH can prevent alterations to the antioxidant defense system in photoaged skin cells. The number of NHDF cells stained with DCFH-DA indicated that intracellular ROS production was higher in the UV + Vehicle-treated group than in the untreated group (No). In contrast, this level was significantly lower in the UV + MEH-treated group compared to the UV + Vehicle-treated group (Figure 2a,b). A similar decrease was detected in the intracellular NO concentration (Figure 2c). SOD activity, SOD expression, and Nrf2 expression showed opposite effects compared to ROS production or NO concentration. SOD activity in UV + MEH-treated cultures increased dose-dependently compared to the UV + Vehicle-treated group (Figure 3). The levels of Nrf2 expression in all UV+MEH-treated groups were enhanced to the level of No group without any difference between the UV + MEH-treated group. On the other hand, the level of SOD expression was increased in only the UV+MMEH-treated group compared to the UV + Vehicle-treated group, while UV + LMEH and UV + HMEH-treated groups were maintained at a constant level (Figure 4). Overall, the high antioxidative activity of MEH can improve the antioxidant ability in dermal fibroblast cells during UV-induced photoaging.

### 3.3. Recovery Effect of MEH on Apoptosis of UV-Irradiated NHDF Cells

Next, this study examined whether MEH effects on the antioxidant ability were accompanied by alterations in UV-induced cell death. The viability of the NHDF cells following UV treatment was reduced to 70% in the UV + Vehicle-treated group compared to untreated cells. The viability of the UV-treated cells cultured with increasing concentrations of MEH was increased in a dose-dependent manner (Figure 5). Furthermore, this study examined whether the improving effects of MEH on the cell viability were associated with inhibition of apoptosis. The total number of apoptotic cells was remarkably higher in the UV + Vehicle-treated group than the untreated group (No), with a concomitant 25% decrease in the total number of live cells. The number of apoptotic cells in the UV + MEH-treated cultures decreased significantly in a dose-dependent manner with a concomitant increase in the number of total live cells (Figure 6a). The expression level of apoptotic proteins was measured in the same sample to verify the mechanism of apoptosis. The expression of the marker proteins responsible for regulating apoptosis was reflected in the changes in the number of apoptotic cells. The increased levels of Bax/Bcl2 expression and the cleavage of Cas-3 due to UV-irradiation were decreased dramatically with the MEH treatment in a dose-dependent manner (Figure 6b). These results suggest that the effects of MEH on antioxidant ability may be associated with the inhibition of UV-induced apoptosis of NHDF cells.

### 3.4. Effect of MEH on the Regulation of ECM Structural Proteins in UV-Irradiated NHDF Cells

UV exposure was tightly linked to the decrease in the abundance of core structural ECM components such as collagens, proteoglycans, elastin, and cell-binding glycoproteins, representing a decrease in ECM structural integrity [23]. The expression levels of MMPs, the activity of intracellular elastase, and the mRNA expression of HAs and hyaluronidases (HAase) were measured in UV + MEH-treated NHDF cells to determine if the effect of MEH on the antioxidant defense system was associated with changes in the UV-induced abnormal structure of the ECM. The levels of MMP-2 and MMP-9 expression, which are collagenases, were significantly higher in the UV + Vehicle-treated group. On the other hand, these levels decreased in a dose-dependent manner after the MEH treatment (Figure 7a). A similar decreasing pattern was observed in the intracellular elastase activity and the mRNA expression of HAase (Figure 7b and Figure 8). These HAase 1 and 2 levels decreased in the UV + MEH-treated cultures compared to the UV + Vehicle-treated group (Figure 8c,d). On the other hand, the mRNA expressions of hyaluronic acid synthases (HAs) 1 and 2 increased in the UV + MEH-treated groups (Figure 8a,b). These results suggest that MEH can prevent the alteration on UV-induced abnormal ECM structures in NHDF cells by inhibiting the levels of collagenase, elastase, and hyaluronidase.

### 3.5. Role of MAPK Signaling Pathway during the Inhibition of UV-Induced Overexpression of MMPs in NHDF Cells

The overproduction of ROS activates the AP-1 by activating the MAPK pathway, resulting in the overexpression of MMPs and the prevention of procollagen synthesis [24]. Therefore, this study analyzed the activation of the MAPK signaling pathway to determine if the inhibitory effects of MEH on regulating MMPs expression are accompanied by inhibition of the MAPK signaling pathway. The ERK, JNK, and p38 phosphorylation levels were elevated significantly in the UV + Vehicle-treated group. On the other hand, these levels were significantly lower in the three UV + MEH-treated cultures, even though the highest level was detected in the UV + HMEH-treated group (Figure 9). These results suggest that the inhibition of the MAPK signaling pathway can be involved the suppression of MMPs overexpression in the UV-induced NHDF cells after the MEH treatment.

### 3.6. Effect of MEH on Inflammatory Response in UV-Irradiated NHDF Cells

UV-induced skin inflammation, including the iNOS-induced COX-2 mediated pathway and NLR family pyrin domain containing 3 (NLRP3) inflammasome, is accompanied mainly by photoaging that worsens the lesions [25]. The changes in the expression of the key regulators in the iNOS-induced COX-2 mediated pathway and inflammasome complex in UV + MEH-treated NHDF cells were analyzed to determine if the improving effect of MEH on the antioxidant defense system was accompanied by a regulation of the UV-induced inflammatory response in dermal fibroblast cells. The levels of iNOS and COX-2 expression were decreased significantly in the UV + MEH-treated cultures, whereas higher levels of these proteins were detected in the UV + Vehicle-treated group (Figure 10). A similar response was observed in the regulation of inflammasome activation. The expression levels of NLRP3, an apoptosis-associated speck-like protein containing a CARD (ASC), and the cleavage of Cas-1 proteins were higher in the UV + Vehicle-treated culture than in the untreated group. On the other hand, these levels were decreased remarkably in the UV + MEH-treated cultures in a dose-dependent manner (Figure 11). Furthermore, the inflammatory cytokines showed a similar pattern in the UV-irradiated NHDF cells after the MEH treatment. The mRNA levels of TNF-α, IL-6, IL-1β, and NF-κB were decreased significantly in a dose-dependent manner in the UV + MEH-treated cultures than in the UV + Vehicle-treated group (Figure 12). Hence, the improvement effect of MEH on the antioxidant defense system may be associated with inhibition of the inflammatory response in the UV-irradiated NHDF cells by regulating the iNOS-induced COX-2 mediated pathway and NLRP3 inflammasome activation.

### 3.7. Effect of MEH on the Melanin Contents in α-Melanocyte-Stimulating Hormone (MSH)-Stimulated B16F1 Cells

Hyperpigmentation is a prominent lesion of photoaged skin [26]. Therefore, this study examined whether the high antioxidant activity of MEH affects the regulation of the melanin content in melanocytes. Accordingly, the changes in the melanin content, tyrosinase activity, and the mRNA expression of melanocyte-inducing transcription factor (MITF), tyrosinase (TYR), tyrosinase-related protein (TYRP)1, and TYRP2 were measured in α-MSH stimulated B16F1 melanocytes after a MEH treatment. The amount of accumulated melanin was 2.5 times higher in the α-MSH + Vehicle-treated group than in the No group. On the other hand, these levels decreased significantly in a dose-dependent manner in the α-MSH + MEH-treated cultures compared to the α-MSH + Vehicle-treated group (Figure 13a). Similar results were observed regarding the tyrosinase activity and the mRNA expression of MITF, TYR, TYRP1, and TYRP2. These levels decreased in the α-MSH + MEH-treated group in a dose-dependent manner (Figure 13b,c). Overall, these results suggest that the high antioxidant activity of MEH may be associated with the prevention of enhancing melanin content in the α-MSH-stimulated melanocytes.

## 4. Discussion

Antioxidant activity is an essential requirement for antiphotoaging drugs because the UV-induced overproduction of ROS participates in the initial stages of cellular damage in the skin [7]. Therefore, natural products with strong antioxidative activity without significant adverse effects have attracted considerable attention as therapeutic strategies to treat photoaging [27]. As part of an ongoing study to discover natural products with high antioxidant activity and antiphotoaging effects, this study investigated the antiphotoaging effects of *H. erecta* in UV-irradiated NHDF cells and α-MSH-stimulated B16F1 cells. The results indicate the mechanisms contributing to the antiphotoaging effects of MEH involving regulation of oxidative stress, apoptosis, the inflammatory response, and melanin content. Furthermore, these results suggest MEH as a treatment for other chronic diseases associated with oxidative stress. On the other hand, further studies will be needed to confirm these antiphotoaging effects in vivo because this study was only conducted in cells derived from the skin.

*H. erecta* is a perennial herb distributed widely in tropical Asia and Indonesia and used as a traditional medicine in the local region [28]. Despite this, there is insufficient scientific evidence for the efficacy of *H. erecta*, with only a few functions being examined in the neurobiology and immunology fields. An n-hexacosanol isolated from *H. erecta* induces the development of central neurons by enhancing the activities of two neuron-specific enzymes [29]. The peripheral administration of n-hexacosanol attenuates the degeneration of septal cholinergic neurons following cerebral injury in rats [30]. Furthermore, n-hexacosanol affects the phagocytic activity, including the morphology, proliferation, protein synthesis, and phagocytosis in murine peritoneal macrophages [31]. The present study demonstrated antiphotoaging effects and their mechanism of action induced by *H. erecta* in dermal cell lines. The results suggest a new function of *H. erecta* that has not been investigated previously.

Thus far, there are very few bioactive compounds identified from the whole plant of *H. erecta*. Among them, verbascoside was first identified from different callus lines and the leaves of *H. erecta* [28]. In addition, several secondary metabolites were isolated from the whole plant of *H. erecta* collected from Jahangirnagar, Savar, Bangladesh. They were revealed as 4-methoxybenzoic acid and 4-methoxycinnamic acid (HE-167), methyl linoleate, methyl stearate (HE-57), Lupeol (HE-100), β-Sytosterol, and Stigmasterol (HE-130) [32]. This study first identified four compounds, including 4-methoxycinnamic acid, 4-methoxybenzoic acid, methyl linoleate, and asterriquinone C-1, in MEH using LC–MS. 4-Methoxycinnamic acid is a cinnamic acid derivative that has been reported to inhibit the diphenolase activity of mushroom tyrosinase effectively [33]. This compound has antioxidant properties because a methyl group (O-CH_3_) in the phenyl ring stabilizes neutral phenoxyl radicals [34]. In addition, methyl linoleate is a fatty acid methyl ester of linoleic acid. This compound has potential therapeutic effects in atherosclerosis, cancer, and some immune dysfunction [35,36]. The methyl ester from vegetable oils of soybean, corn, and sunflower has stronger antifungal and antioxidant activity against *Paracoccidioides spp.* and *Candida spp.* [37]. The potential antibiotic, antifungal, antiproliferative, tyrosinase inhibitory activity, antioxidant, antidiabetic, and cytotoxic effects of 4-methoxybenzoic acid were investigated [38]. Among these activities, stronger antioxidant activity was attributed to a dihydroxyl group bonded to the ortho position by improving the stability of aryl oxyl radicals [39,40]. Baicalin (5,6-dihydroxy-7-O-glucuronide flavone) is also a flavonoid that has been investigated for its antiviral activity against dengue virus [41]. Furthermore, asterriquinone C-1 was detected as the main component of MEH based on LC–MS analysis results. This compound has high antioxidant activity because the indol ring contributes to the high resonance stability of the free radical scavenging activity for ROS and reactive nitrogen species (RNS), as well as a low energy barrier [42,43]. Therefore, the antioxidant activity of MEH is related to the structure–activity relationship of the above compounds.

ROS overproduction during UV irradiation induces significant alterations in the antioxidant defense system and causes cell death [27]. ROS overproduction can destroy the balance between ROS production and antioxidant defenses, which decreases the activities of antioxidant enzymes [44]. Oxidative stress, caused by the inhibition of antioxidant enzymes, induces the formation of lipid peroxidation, which leads to a disruption of the cell membrane and cell death [44]. Thus far, several natural products with high radical scavenging activity have been shown to effectively restore the antioxidant defense system and cell death in UV-irradiated cells [44,45,46]. Fuzhuan-brick tea (FBTA) exhibits high ROS scavenging activity for various types of radicals, including ABTS and DPPH. Moreover, FBTA activates heme oxygenase 1 and SOD by upregulating Nrf2, thereby suppressing UV-induced cytotoxicity at a dose of 100 μg/mL in UV-irradiated HaCaT cells [45]. Caffeic acid and sinapic acid reduce UVB-induced intracellular ROS production and reduce UVB-induced cell death by up to 95.98% and 91.9%, respectively, with a high antioxidant effect [46]. *Viris vinifera* L. water extract, which is rich in polyphenols, suppresses UV-induced cell death by regulating apoptosis and necrosis, decreases the sub G1 population, and relieves cell cycle arrest [47]. In the current study, the antioxidative effects of MEH were investigated by measuring the intracellular ROS levels, NO concentration, and expression of the Nrf2 pathway to demonstrate MEH as a candidate antiphotoaging drug. Consistent with the other natural products described above, MEH suppressed the intracellular ROS levels and NO concentration associated with the upregulation of Nrf2 and SOD. Furthermore, MEH reduced UV-induced cell death in a dose-dependent manner, which was associated with the regulation of apoptosis, including suppression of Bax/Bcl2 levels and cleavage of Caspase-3. These results suggest that MEH suppresses UV-induced cell death by regulating apoptosis with its high antioxidative activity, highlighting its potential as an antiphotoaging drug.

Deep wrinkles, which are one of the most prominent symptoms of photoaging, are formed by activating collagenases and elastases in the skin after UV irradiation [48]. During this process, the degradation of HA is accelerated by the UV-induced downregulation of HAs expression and the upregulation of HAase [49]. Some natural products effectively improve wrinkle formation due to UV irradiation. Rb-ME has been shown to attenuate MMP-9 gene expression and elevate HA2 gene expression by inhibiting p38 phosphorylation and inactivating AP-1 [15]. Wheat extract oil (WEO) inhibited UVB-induced changes in tissue procollagen type I, HA, and ceramide [16]. UMH reduced the expression of MMPs, which led to the inhibition of collagen degradation. In addition, the oral administration of the UMH extracts decreased the depth, thickness, and length of wrinkles on UVB exposed hairless mice [17]. The present study analyzed the effects of MEH on ECM structure by measuring the expression of MMPs and the activation of elastase, in addition to the expression of HAs and HAase. The results using *H. erecta* extracts were similar to those of previously reported extracts, but the study further examined the regulation of the associated signaling pathway. However, the UV + MEH-treated group showed lower expression level of MMP9 proteins than untreated group. This result is believed to be responsible for suppressing MMP9 expression or stimulating MMP9 degradation by certain components in MEH, although further research will be needed.

Because UV-irradiation induces the expression of pro-inflammation-related genes, the inflammatory response acts as an important mediator of photoaging [50]. The generated ROS, damaged DNA, and altered cell homeostasis caused by UV photons activate the inflammasome, releasing pro-inflammatory cytokines (e.g., IL-1β and IL-18) to form a local microenvironment for inflammation [51,52]. In addition, two key enzymes, iNOS and COX-2, are activated to overexpress superoxide and NO radicals that mediate inflammation [53,54]. In the present study, the anti-inflammatory activity of MEH was determined based on the correlation between UV-induced inflammation and antioxidative activity. MEH suppressed UV-induced inflammation by diminishing iNOS, COX-2 derived inflammation, activation of the inflammasome, and the expression of inflammatory cytokines. Similar responses were detected in UV-irradiated cells after treatment with several natural products. Among them, a red raspberry extract had a suppressive effect on the UVB-induced inflammatory cascade, including c-jun and attenuated activation of NF-κB and COX-2 [55]. MED exhibited an anti-inflammatory effect on UV-irradiated NHDF cells by suppressing the iNOS-induced COX-2 mediated pathway and inflammasome activation [19]. *Rhus javanica* extract (RJE) had preventive effects against UVB-induced expression and activation of COX-2, MMP-1, and MMP-13 [56].

Along with the ECM alteration, hyperpigmentation is a major symptom of photoaging, caused by the UV-induced accumulation of melanin pigments in the skin [57]. UV-irradiation stimulates keratinocytes to release α-MSH, which upregulates the expression of MITF in melanocytes [57]. This response stimulates the upregulation of TYR and TYRPs expression to produce and accumulate melanin pigment [57]. The hyperpigmentation by melanin was also suppressed after treatment with several natural products. Rb-ME inhibited melanogenesis by suppressing tyrosinase, MITF, and TYRP-1 mRNA in B16F10 cells following α-MSH treatment [15]. The hot water extract of *Pleurochrysis carterae* inhibited melanin synthesis by downregulating the tyrosinase and MITF levels in B16F1 melanoma cells [58]. A Hoelen extract repressed melanin synthesis by inhibiting TYRP2 gene transcription while the tyrosinase expression remained constant [59]. The white rose petal extracts (WRPE) effectively inhibited the activity of tyrosinase, while *Eriobotrya japonica* leaf ethanol extract (EJEE) suppressed melanin contents with their strong antioxidative activity [60,61]. The current study measured the suppressive effects of MEH on melanin accumulation in α-MSH-treated B16F1 cells. α-MSH upregulated the melanin content and tyrosinase activity. MEH reversed the expression of MITF, TYR, and TYRP mRNA. These effects are similar to previous efficacy data for other natural products.

## 5. Conclusions

This study examined whether the high antioxidant activity of MEH was associated with the reduced activation of UV-induced photoaging pathways in skin fibroblast and melanoma cells. The current study showed that the MEH treatment successfully prevented the apoptosis, inflammatory response, ECM modulation, and hyperpigmentation in UV-irradiated NHDF and α-MSH-treated B16F1 cells by improving the antioxidant defense system. These effects also highlight the potential application of MEH as a candidate for photoaging treatment. However, our study had some limitations in that it did not comprehensively investigate the activity of any of the pathways potentially regulated by MEH, only the expression of one or a few members of each pathway and only at a single timepoint due to the fact that UV-induced photoaging pathways are much more complex. Moreover, the lack of any comparison in relation to other human dermal fibroblasts and melanoma cell lines should be considered as a drawback of our study.

## Figures and Tables

**Figure 1 antioxidants-11-01317-f001:**
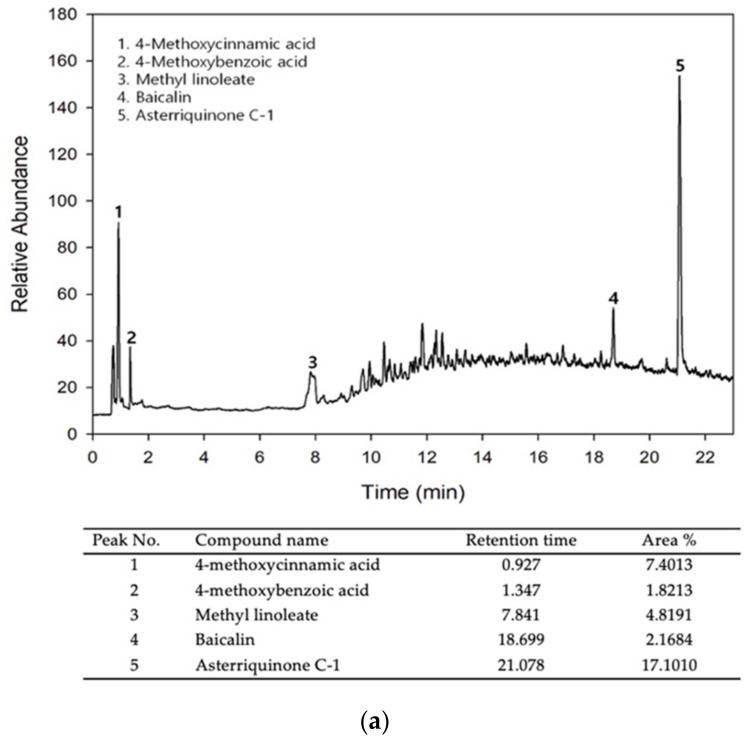
LC–MS analysis and DPPH scavenging activity of MEH. (**a**) Chromatogram of MEH. Five active components, (**1**) 4-methoxycinnamic acid, (**2**) 4-methoxybenzonic acid, (**3**) methyl linoleate, (**4**) baicalin, and (**5**) asterriquinone C-1, were detected in the chromatogram; (**b**) Chemical structure of the five bioactive components detected in Figure 1a; (**c**) DPPH radical scavenging activity of MEH. Three MEH dry samples were used to prepare the solution of MEH, and scavenging activity was measured in triplicate. Data are reported as the mean ±SD. Abbreviations: LC–MS/MS, liquid chromatography–tandem mass spectrometry; DPPH, 2,2-diphenyl-1-picrylhydrazyl; MEH, Methanol extract of *Hygrophila erecta* (Brum. F.) Hochr.

**Figure 2 antioxidants-11-01317-f002:**
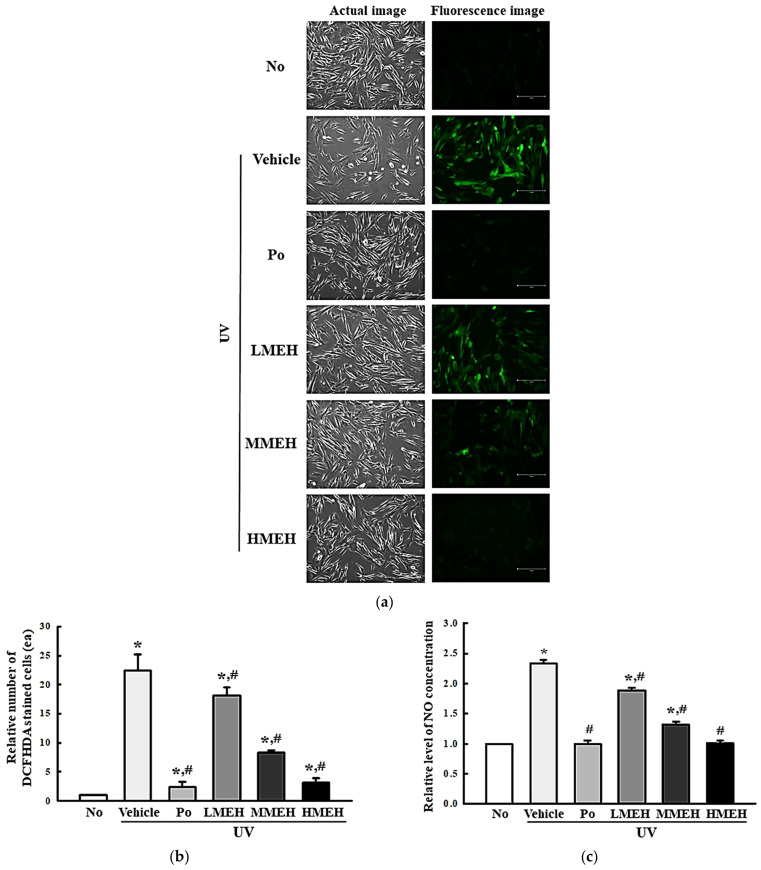
Detection of ROS and NO in normal human dermal fibroblasts. (**a**,**b**) Determination of intracellular ROS production. The cells were stained with DCFH-DA, and green fluorescence-stained cells were observed using a fluorescent microscope at 200× magnification; (**c**) Determination of the NO concentration. Three wells per group were used for DCFH-DA staining and NO assay, and the assays were repeated for three times. Data represent the mean ± SD of triplicates. *, *p* < 0.05 relative to the untreated group. #, *p* < 0.05 compared to the UV + Vehicle group. Abbreviations: ROS, reactive oxygen species; NO, Nitric oxide; Po, Positive control; DCFH-DA, 2′,7′-Dichlorofluorescin diacetate; MEH, Methanol extract of *Hygrophila erecta* (Brum. F.) Hochr.

**Figure 3 antioxidants-11-01317-f003:**
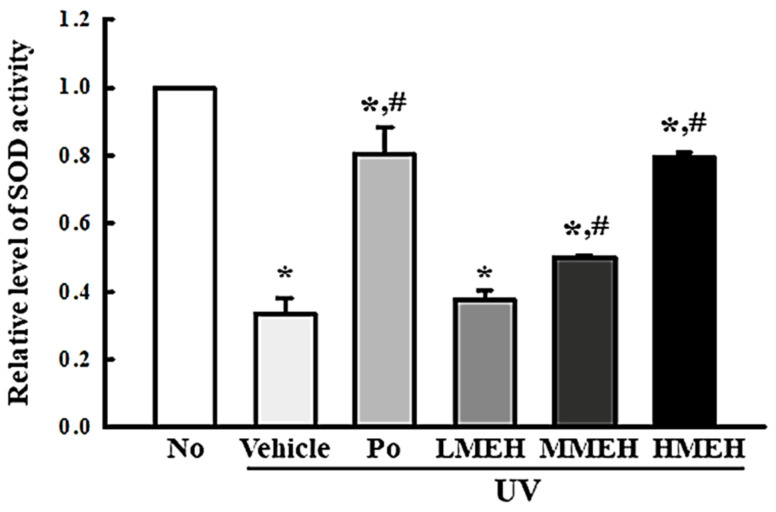
Determination of SOD activity in normal human dermal fibroblasts. One SOD unit is defined as the amount of the enzyme in the sample (20 µL) that inhibits the reduction reaction of water-soluble tetrazolium salt-1 (WST-1) with superoxide anion by 50%. Three to five dishes per group were used to measure SOD activity, and the assay was repeated for three times. The SOD activity was analyzed in triplicate. Data are reported as the mean ±SD of triplicates. *, *p* < 0.05 compared with the untreated group. #, *p* < 0.05 compared with the UV + Vehicle group. Abbreviations: SOD, Superoxide dismutase; WST-1, Water-Soluble Tetrazolium 1; Po, Positive control; MEH, Methanol extract of *Hygrophila erecta* (Brum. F.) Hochr.

**Figure 4 antioxidants-11-01317-f004:**
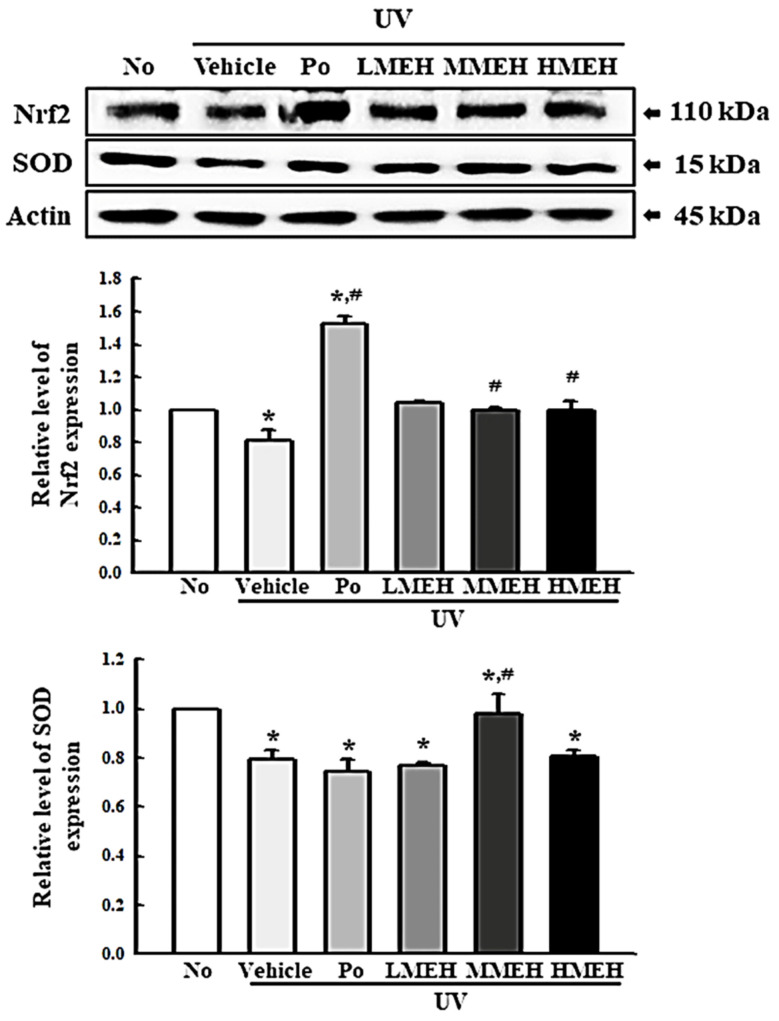
Detection of SOD and Nrf2 protein expression in normal human dermal fibroblasts. Expression level of the proteins was detected with the specific antibodies and quantified using an imaging densitometer. Determination of SOD activity. One SOD unit is defined as the amount of the enzyme in the sample (20 µL) that inhibits the reduction reaction of water-soluble tetrazolium salt-1 (WST-1) with superoxide anion by 50%. Three dishes per group were used for Western blot assay, and the assay was repeated fir three times. Data are reported as the mean ± SD of triplicate determinations. *, *p* < 0.05 compared with the untreated group. #, *p* < 0.05 compared with the UV + Vehicle group. Abbreviations: SOD, superoxide dismutase; Nrf2, nuclear factor erythroid 2–related factor 2; Po, Positive control; MEH, Methanol extract of *Hygrophila erecta* (Brum. F.) Hochr.

**Figure 5 antioxidants-11-01317-f005:**
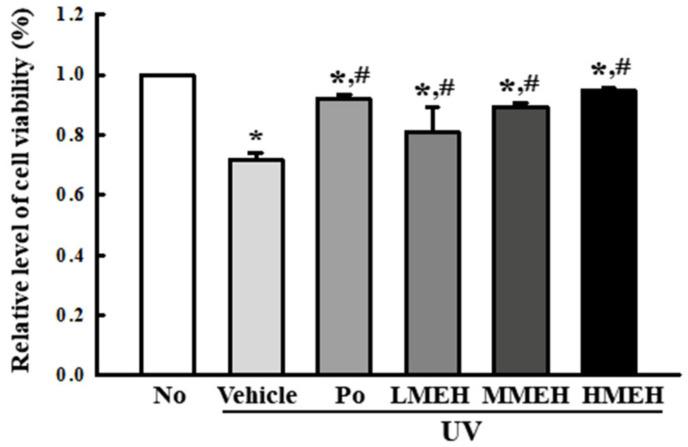
Cytotoxicity of UV+MEH-treated NHDF cells in normal human dermal fibroblasts. The optical density of MTT assay was measured in triplicates and the assay was repeated for three times. Data are reported as means ±SD. *, *p* < 0.05 relative to the untreated group. #, *p* < 0.05 compared to the UV + Vehicle group. Abbreviations: MTT, 3-(4,5-dimethylthiazol-2-yl)-2,5-diphenyltetrazolium Bromide; Po, Positive control; MEH, Methanol extract of *Hygrophila erecta* (Brum. F.) Hochr.

**Figure 6 antioxidants-11-01317-f006:**
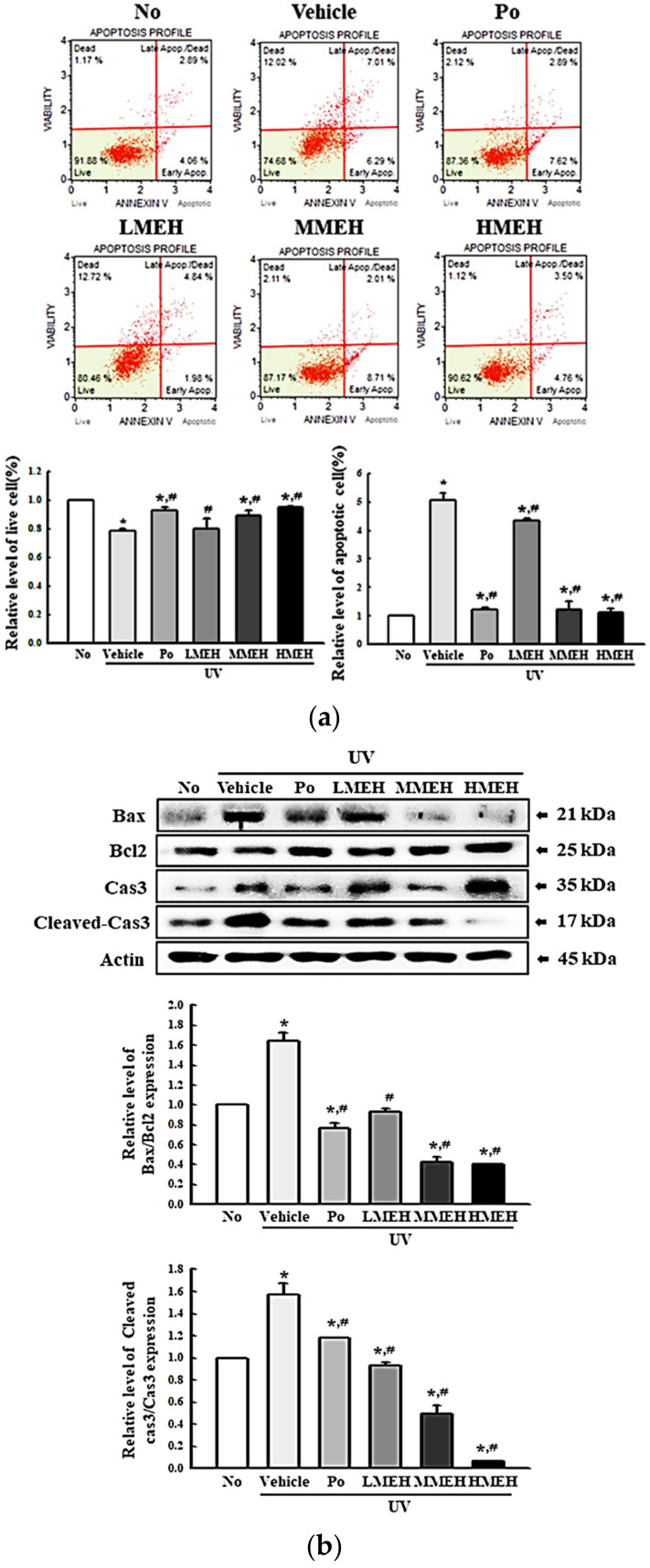
Apoptosis related parameters analysis in normal human dermal fibroblasts. (**a**) Determination of live and apoptotic cells. Two to three wells per group were used for Annexin V and 7-AAD staining. The population analysis was performed in triplicate and the assay was repeated for three times; (**b**) Expression levels of apoptotic proteins. Bands for four apoptotic proteins including Bax, Bcl2, Cas-3 and Cleaved Cas-3 on the membrane were detected with the specific antibodies and quantified using an imaging densitometer. Three dishes per group were used to prepare the cell lysates. Western blot analysis was performed in triplicate and repeated for three times. The data are reported as the mean ±SD of triplicates determination. *, *p* < 0.05 compared to the untreated group. #, *p* < 0.05 compared to the UV + Vehicle group. Abbreviations: 7-AAD, 7-Aminoactinomycin D; Cas-3, Caspase 3; Bax, Bcl2-associated X protein; Bcl2, B-cell lymphoma 2; Po, Positive control; MEH, Methanol extract of *Hygrophila erecta* (Brum. F.) Hochr; IgG, Immunoglobulin G.

**Figure 7 antioxidants-11-01317-f007:**
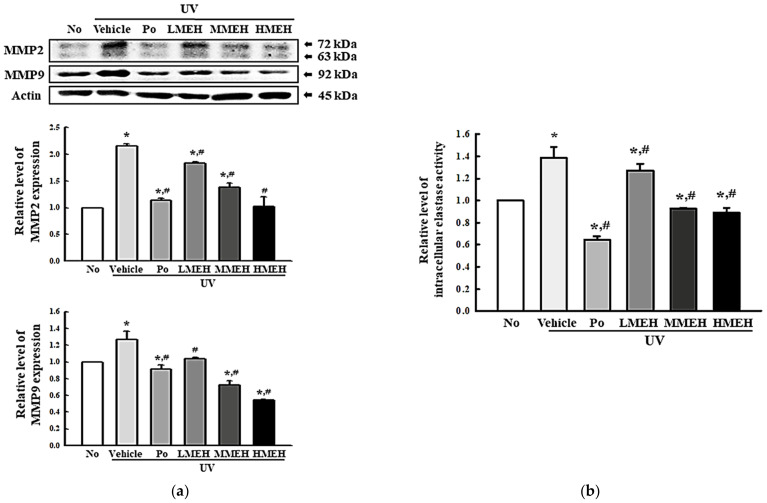
Detection of MMPs and elastase in normal human dermal fibroblasts. (**a**) Levels of MMP-2 and MMP-9. Three dishes per group were used to prepare the cell lysates. Western blot analysis was performed in triplicate and repeated for three times. The relative level of elastase activity (**b**) was determined in total cell lysate using the substrate. Three dishes per group were used to prepare the cell lysates, and the elastase activity assay was performed in triplicate. This experiment was performed for three times and results are the average of all experiments. The data are reported as the means ±SD. * *p* < 0.05 compared to the untreated group. # *p* < 0.05 compared to the UV + Vehicle group. Abbreviations: MMP, Matrix metalloproteinase; Po, Positive control; MEH, Methanol extract of *Hygrophila erecta* (Brum. F.) Hochr.

**Figure 8 antioxidants-11-01317-f008:**
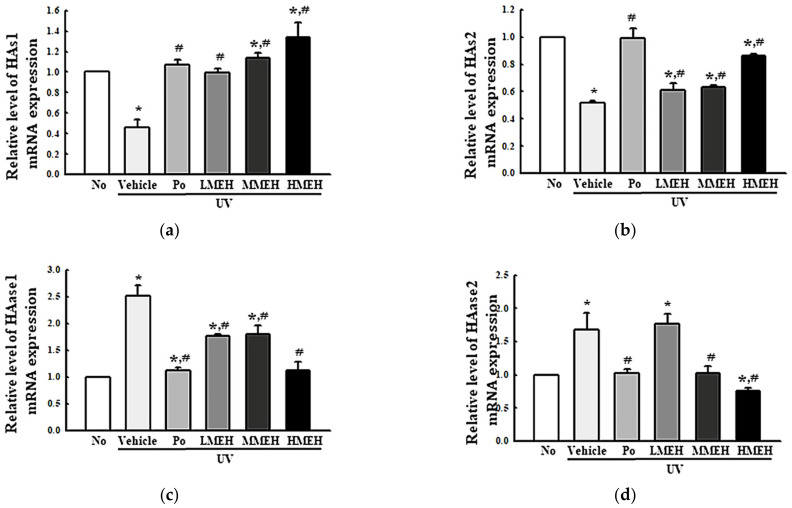
Expression of the ECM components in normal human dermal fibroblasts. The expression levels of HAs (**a**,**b**) and HAase (**c**,**d**) mRNA were measured in the total RNA prepared from three dishes per group after treatment of each concentration of MEH. RT-qPCR analysis was performed in triplicate. This experiment was performed for three times and results are the average of all experiments. The data are reported as the means ± SD of triplicate determinations. * *p* < 0.05 compared to the untreated group. # *p* < 0.05 compared to the UV + Vehicle group. Abbreviations: ECM, Extracellular Matrix; HAs, Hyaluronic Acid synthase; HAase, Hyaluronic Acid lyase; RT-qPCR, Quantitative reverse transcription PCR; Po, Positive control; MEH, Methanol extract of *Hygrophila erecta* (Brum. F.) Hochr.

**Figure 9 antioxidants-11-01317-f009:**
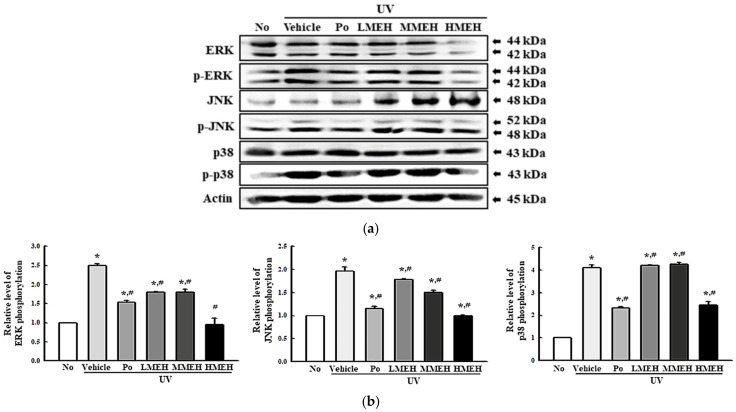
Expression of the key members in the MAPK pathway in normal human dermal fibroblasts. (**a**) Image of western blot. (**b**) Relative level of each protein. Three dishes per group were used to prepare the cell lysates. Western blot was performed in triplicate. This experiment was performed for three times and results are the average of all experiments. The data are reported as the means ±SD of triplicate determinations. *, *p* < 0.05 compared to the untreated group. #, *p* < 0.05 compared with the UV + Vehicle group. Abbreviations: MAPK, Mitogen-activated protein kinases; ERK, Extracellular-signal-regulated kinase; JNK, Jun N-terminal kinase; Positive control; MEH, Methanol extracts of *Hygrophila erecta* (Brum. F.) Hochr.

**Figure 10 antioxidants-11-01317-f010:**
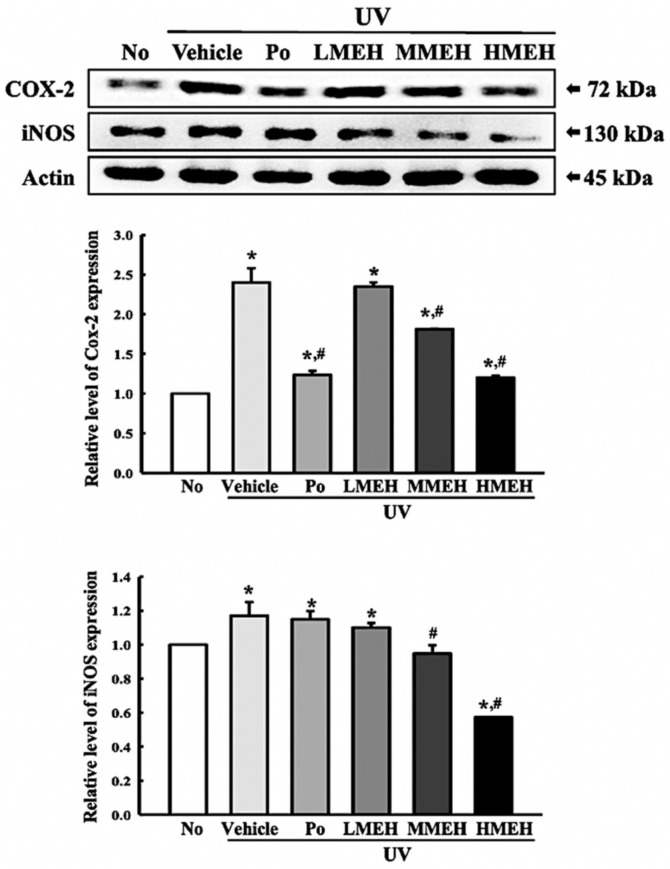
Expression of COX-2 and iNOS proteins in normal human dermal fibroblasts. Three dishes per group were used to prepare the cell lysates. Western blot was performed in triplicate. This experiment was performed for three times and results are the average of all experiments. The data are reported as the means ±SD of triplicate determinations. *, *p* < 0.05 compared to the untreated group. #, *p* < 0.05 compared to the UV + Vehicle. Abbreviations: COX-2, Cyclooxygenase-2; iNOS, Inducible nitric oxide synthase; Po, Positive control; MEH, Methanol extracts of *Hygrophila erecta* (Brum. F.) Hochr.

**Figure 11 antioxidants-11-01317-f011:**
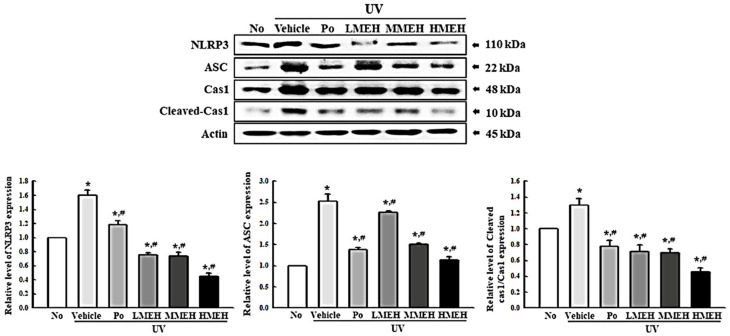
Expression of inflammasome proteins in normal human dermal fibroblasts. Three dishes per group were used to prepare cell lysates. Western blot was performed in triplicate. This experiment was performed for three times and results are the average of all experiments. The data are reported as the means ±SD of triplicate determinations. *, *p* < 0.05 compared to the untreated group. #, *p* < 0.05 compared to the UV + Vehicle group. Abbreviations: NLRP3, NLR family pyrin domain containing 3; ASC, Apoptosis-associated speck-like protein; Cas-1, Caspase 1; IgG, Immunoglobulin G; Po, Positive control; MEH, Methanol extract of *Hygrophila erecta* (Brum. F.) Hochr.

**Figure 12 antioxidants-11-01317-f012:**
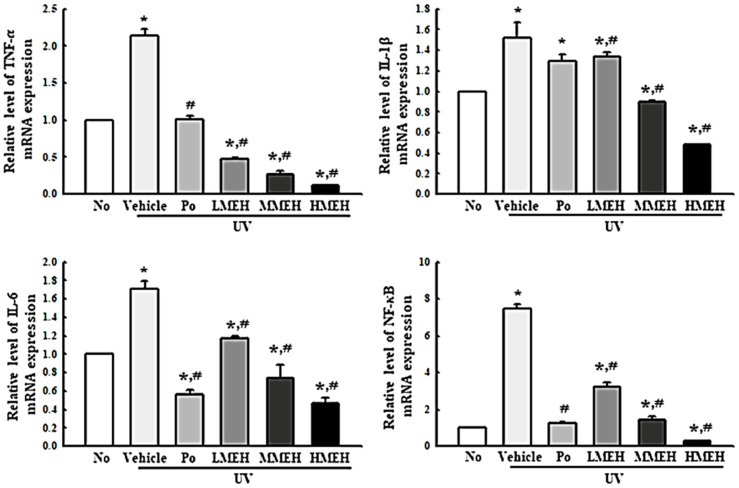
Level of the inflammatory cytokines in normal human dermal fibroblasts. Three dishes per group were used to prepare the total RNA. RT-qPCR analysis was performed in triplicate. This experiment was performed for three times and results are the average of all experiments. The data are reported as the means ±SD of triplicate. *, *p* < 0.05 compared to the untreated group. #, *p* < 0.05 compared to the UV + Vehicle group. Abbreviations: TNF-α, tumor necrosis factor α; IL-6, Interleukin 6; IL-1β, Interleukin 1β; NF-κB, Nuclear factor kappa light chain enhancer of activated B cells; UV, Ultraviolet; RT-qPCR, Quantitative reverse transcription PCR; Po, Positive control; MEH, Methanol extract of *Hygrophila erecta* (Brum. F.) Hochr.

**Figure 13 antioxidants-11-01317-f013:**
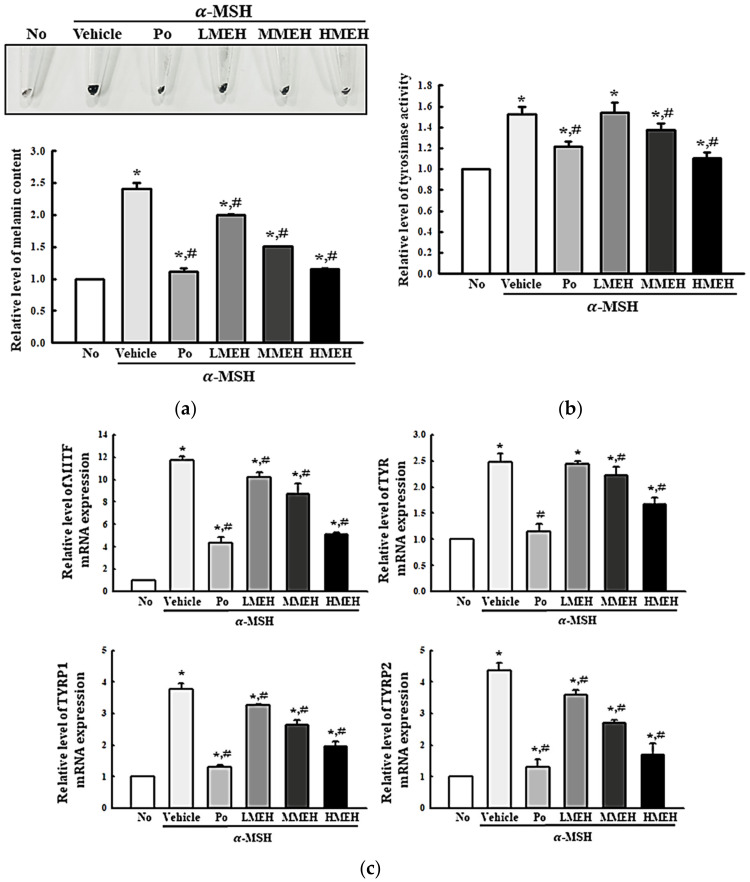
Regulation of melanin synthesis in B16F1 melanoma cells. (**a**) Melanin content analysis; (**b**) After preparing the cell lysates, the tyrosinase activity was measured with L-DOPA. Three to five dishes per group were used to prepare the cell lysates in duplicate. The melanin content was measured in triplicate. This experiment was performed for three times and results are the average of all experiments; (**c**) The levels of MITF, TYR, TYRP1, and TYRP2 mRNA were measured by RT-qPCR. Three dishes per group were used to prepare the total RNAs. RT-qPCR was performed in triplicate. This experiment was performed for three times and results are the average of all experiments. Data are reported as the means ±SD of triplicate. *, *p* < 0.05 compared to the untreated group. #, *p* < 0.05 compared to the UV + Vehicle group. Abbreviations: UV, Ultraviolet; RT-qPCR, Quantitative reverse transcription PCR; α-MSH, alpha-Melanocyte-stimulating hormone; L-DOPA, 3,4-Dihydroxy-L-phenylalanine; MITF, Melanocyte Inducing Transcription Factor; TYR, Tyrosinase; TYRP, Tyrosinase related protein; Po, Positive control; MEH, Methanol extract of *Hygrophila erecta* (Brum. F.) Hochr.

## Data Availability

All the data that support the findings of this study are available on request from the corresponding author.

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
