# Peer review of "Antioxidative Role of Hygrophila erecta (Brum. F.) Hochr. on UV-Induced Photoaging of Dermal Fibroblasts and Melanoma Cells"

_antioxidants, 2022, doi:10.3390/antiox11071317_

Round 1
Reviewer 1 Report
The authors revised and extended the manuscript.
Author Response
According to your comments, we have corrected our manuscripts entitled “Antioxidative role of Hygrophila erecta(Brum. F.) Hochr. on UV-induced photoaging of dermal fibroblasts and melanoma cells”. The corrected Manuscript and Response to comments have been attached to this letter. Thank you very much for your cooperation.
Reviewer 2 Report
The authors have made considerable improvements to their manuscript. However, the number of amendments required is still very high. I feel that many of the corrections are related to English language issues and it is possible that the English language service that the authors have used does not have the ability to discern where the phrases and sentences that they are correcting grammatically do not accurately represent the results that the authors are depicting in their figures.
1. English language editing and removal of repetition of information has improved the quality of parts of the manuscript and the presentation of the experiments and results. Final English language is required in some places (e.g. ‘these cell deaths were caused by apoptosis’ (should be ‘cell death was caused by apoptosis’)) and revision of phrases that are grammatically incorrect (e.g. “the improving effect of MEH on the antioxidant defence system”) will be required prior to final publication. The authors should also carefully check spelling in figures and figure legends. For example, in the legend for Figure 8, HAs is incorrectly written as Has.
2. There is still some confusion about how many times experiments were performed in this study. Each time that an experiment is performed, it is expected that there are replicates for each treatment group. For example, if cells are being treated with a particular concentration of a chemical, then there might be 3-5 wells or flasks that are treated with each concentration of the chemical. In addition, each experiment must be performed multiple times (e.g. 3 times). This means that on 3 different occasions, the same experiment (with triplicate wells for each treatment group) is set up and analysed, and in reporting results for their manuscript, each of the experiments should have shown similar results (e.g. growth stimulation or inhibition, etc). The authors have now included information that triplicate or duplicate wells were included for each treatment group in each experiment. However the number of (independent) times that each experiment was performed does not seem to be included. This requires clarification.
3. In each of the figure legends, the cell line used to produce the results that are included in that figure should be stated. For example in Figure 2: “Detection of ROS and NO in normal human dermal fibroblasts.”
4. Page 11, line 373: The authors state that viability of the NHDF cells was less than 30% following UV treatment, however in Figure 5, it appears that viability was 70% (that is, viability was reduced by ~30%). This requires correction.
5. Page 11, line 378: A similar mistake has been made here. Based on the image in Figure 6a (graph on the left), the “decrease in live cells” appears to be 25%, not 75%. This requires correction.
6. In lines 386 and 387, use of the words ‘improved’ and ‘improvement’ appears to be inappropriate. In line 386, ‘improved’ can be removed as it is redundant in this context. In line 386, “reduced UV-induced apoptosis” is a more accurate description of results. In line 406, the word ‘improved’ could also be removed.
7. The figure legend for Figure 6 contains a number of errors. It is difficult to read and follow as wording that has been deleted is still present and the sentences do not seem to make sense when the deleted words are removed.
8. Repetition of Western blotting methods details should be removed from the legends to Figures 7 - 13. It is not necessary to state methods details such as that proteins are detected with specific antibodies or quantified using a densitometer, etc. General methods that are applicable every time the method was performed in the study, for example if experiments included triplicate wells for each treatment group and that each experiment was performed three times, should just be stated in the Methods section. However, numbers of replicates should be included in the relevant figure legend(s) if they are not identical to the general method described in the Methods section. If experiments were performed three times and representative results from one of those experiments, or alternatively, if experiments were performed three times and results are the average of all three of the experiments, that type of detail should be stated in the figure legend.
9. In line 406, the authors state “the UV-induced abnormal structure of the ECM”, however they have not mentioned previously that ECM structure in UV-treated cells is expected to be abnormal. It is not clear whether this is another English language error that hasn’t been picked up by the English language editor and the authors are investigating ECM changes in their UV-treated cells, or whether an introductory sentence is required to explain why they expect ECM changes (e.g. previously published reports). Without prior information, the conclusions of this section (lines 415-417) do not seem to be justified.
10. MEH treatment of UV-treated cells seems to reduce MMP-9 levels to levels that are markedly lower than MMP-9 expression in untreated cells. Do the authors have an explanation for this? (A similar pattern of results is also evident for several additional factors investigated in MEH-treated cells).
11. The concluding statement in section 3.5 (lines 443-445) does not seem to be justified based on the Western blots presented in Figure 9. The authors have provided no causal link between MAPK signalling and MMP expression in these cells. It is suggested that the statement is modified to more appropriately reflect the nature of the experiment performed.
12. In this study, the authors treat the cells with MEH immediately after the cells are irradiated with UVB. Due to this study design, it seems the MEH is preventing UV-induced changes in the expression of factors that the authors are measuring (e.g. inflammatory cytokines, etc) rather than reversing changes that UV irradiation has caused. If this is the case, descriptions and interpretation of MEH effects throughout the manuscript should always reflect this. In order to demonstrate that MEH facilitates recovery from UV-induced damage to cells, a different study design would have to be employed.
13. It is suggested that the authors add a few sentences or a short paragraph to the Discussion section in which they summarise the limitations of the present study. A notable limitation is that the authors have not comprehensively examined the activity of any of the pathways potentially regulated by MEH, only the expression of one or a few members of each pathway and only at a single timepoint. Use of only a single line of normal human dermal fibroblasts and a single melanoma cell line forms another limitation of the work. Inclusion of these and other limitations of the work will more appropriately present the study findings and clearly define future studies that build upon results generated in this series of investigations.
Author Response

(The authors gave the same response as above.)

Reviewer 3 Report
Fig1b: It is better to quantify main compostion.
Fig 2b/c, Fig3…… Try to use group comparison for data.
The paper should discuss the structure-activity relationship, especially the main compounds.
Author Response
According to your comments, we have corrected our manuscripts entitled “Antioxidative role of Hygrophila erecta(Brum. F.) Hochr. on UV-induced photoaging of dermal fibroblasts and melanoma cells”. The corrected Manuscript and Response to comments have been attached to this letter. Thank you very much for your cooperation.
This manuscript is a resubmission of an earlier submission. The following is a list of the peer review reports and author responses from that submission.
Round 1
Reviewer 1 Report
The article of Su Jin Lee et al. entitled “Antioxidative role of Hygrophila erecta (Brum. F.) Hochr. on UV-induced photoaging of dermal fibroblasts and melanoma cells” discusses the antioxidant effect of Hygrophila erecta on fibroblasts and melanoma cells from UV exposure. The article is fluid in its drafting and clear in the methodologies applied. However, the authors should implement the introduction. They should better explain the mechanism of action by which ROS alters cells (lines 40-48) and include the extended name for NF-Kb and MMP proteins. In addition, the part on antioxidants could be implemented by inserting a definition of antioxidants and explaining better what vitamin A and fluorouracil are. The bibliography should be updated with more recent articles.
Reviewer 2 Report
This is an interesting manuscript describing the effects of an extract of H. erecta in skin cells. In this preliminary study, the authors have screened multiple pathways but have not examined any in a comprehensive manner. The findings of the research are also interesting, noting that all results are based on a single human dermal fibroblast cell line and a single murine melanoma cell line, with no verification of results in either other cell lines or in vivo using experimental models. The manuscript is quite difficult to read due to English language issues and because the Results text and figure legends repeat large amounts of the methods protocols, which are well-described in the Methods section. There also appears to be a number of errors in the figures and text. Despite issues with the presentation of the manuscript, I feel that it is suitable for publication following amendment and extensive English language editing. It is also possible that some experiments will need to be repeated due to insufficient replicates being performed. Major comments are listed below and I have also added some English language corrections and additional comments to selected parts of the pdf copy of the manuscript. Note that my comments and corrections cover only part of the Results section.
- English language editing of the manuscript text will be required as there are many grammatical and syntax errors. I have corrected a number of errors in the pdf copy, however I have not added corrections to all parts of the Results section. Professional English language assistance prior to submission of an amended manuscript is essential as it is very difficult to understand the authors’ descriptions or interpretation of their results.
- Overall, the text in the Results section is lengthy and very repetitive. As protocols are clearly described in the Methods section, there is no need to repeat the methods in the manuscript text or in the figure legends. Once these extraneous descriptions are removed, the explanations of the findings will become much easier to follow and greater prominence can be given to the important results.
- It is not appropriate to report SD when only duplicates are used. This comment applies to all experiments reported in the study. As it is more acceptable to perform experiments at least 3 times in order to confirm results, the authors should ascertain whether particular experiments need to be repeated.
- The final sentence in the Introduction section should be re-written. Readers will not be able to understand it because prior to this point, no background information has been given about extracts derived from H. erecta and their activity in other cells. Unexpected references to neurons and macrophages are confusing in a manuscript about UV damage and photoaging.
- Line 88, first word: Should this be methanol (not water)?
- What is the control for elastase activity assays? This should be added.
- Line 278: Please confirm that the annealing temperature was 70C. This is quite high for an annealing temperature.
- ALL figure legends (and text): It is not necessary to repeat general methods in either the figure legends or the text. Once these additional sentences are removed from the document, the manuscript will be much easier to read.
- The text is very difficult to follow as the authors have not always stated the effects of UV treatment on each factor, nor have they clearly detailed MEH effects. An example amendment (lines 322-324) would be:- “UV treatment caused a marked decrease in cellular SOD activity, and although this decrease was diminished in a dose-dependent manner in MEH-treated cultures, levels remained significantly less than SOD activity in untreated cells (Figure 3).” It is suggested that the authors go through the text in the Results section to ensure that the descriptions adequately describe and properly match the results presented in the figures.
- The effects of UV + MEH treatment of cells depicted in Figure 4 do not seem to match the descriptions in the text. It is suggested that the text in lines 322-326 is modified to better reflect the results.
- In line 360, the authors state that cell viability was less than 30% in UV-treated cells, however, it appears from the graph in Figure 5 that cell viability was just less than 80%. Could the results and text please be re-checked and amended where necessary?
- The results depicted in Figure 5 and Figure 6a do not seem to match as the number of live cells in UV + Vehicle treated cells is just under 80% in results shown in Figure 5, but 25.83% in Figure 6a. How do the authors explain this discrepancy? (And how do these results compare with result in the Supplementary figures where methods were optimised)?
- There are many instances in the Results text where the authors appear to have over-interpreted their findings. Correlation (co-occurrence) does not imply causation, especially in the context of this manuscript where none of the pathways has been comprehensively examined. These statements need to be modified so that they more accurately reflect both the types of experiments performed and the results of those experiments.
- Supplementary Figure 1 is labelled as ‘Determination of the optimal UV dosage’ however the images and graph labelling seem to depict MEH dosage. Could the authors please clarify the content of the Figure and the text associated with this figure?
- In the manuscript text (lines 133-1135), it appears that Supplementary Figure 3 should depict MEH dose response curves, however it seems to depict UVB dose response. Could this please be clarified?

Reviewer 3 Report
1. Language expression should be improved. Especially, Abstract should be rewritten.
2. According to Fig 1a, there are more than four compounds. Other compounds should be identified.
3. The main compounds in MEH should be quantified.
4. In the discussion, the structure-activity relation should be discussed.